# A Paratext Perspective on the Translation of the *Daodejing*: An Example from the German Translation of Richard Wilhelm

**Xiaoshu Li and Yuan Tan \***

School of Foreign Languages, Huazhong University of Science and Technology, Wuhan 430074, China; d201881124@hust.edu.cn

**\*** Correspondence: tanyuan@hust.edu.cn

**Abstract:** In the German translation history of the *Daodejing*, the version rendered by the renowned German sinologist, Richard Wilhelm, has vigorously propelled the study of Laozegetics in Germany and stands as a translation of historical and scholarly significance. Wilhelm complemented the concise main text through the use of diverse, precise, and appropriate paratexts, granting his translation both readability and academic rigor. This ensures the admiration of general readers and the recognition of professional scholars. Tailored to the linguistic preferences and educational levels of German readers, Wilhelm frequently employed highly recognizable theological, philosophical, and literary concepts within the German cultural system to elucidate the *Daodejing*. This translation strategy effectively satisfies the expectation horizon of target readers. In the paratexts, Wilhelm constructs a philosophical framework of Daoism, compares the thought of Confucianism and Daoism, and broadens the dialogue between Chinese philosophical thought and Western intellectual traditions, thereby bestowing upon the *Daodejing* a renewed vitality in the German-speaking world.

**Keywords:** *Daodejing* 道德經; Richard Wilhelm; paratext; Laozegetics

## 1. Introduction

The *Daodejing*, frequently referred to as *Laozi*, occupies a significant position within the corpus of Chinese philosophical classics. As the cornerstone text of Daoism, it is recognized globally as crucial for studies in Chinese philosophy, culture, religion, and history. As noted by Misha Tadd, the number of translations and commentaries on the *Daodejing* is massive, boasting a remarkable 2051 versions in 97 different languages (Tadd 2022b). This places the *Daodejing* in the most extensively translated and deeply impactful literary compositions worldwide. Contemporary academic discourse on the *Daodejing*'s translations mainly emphasizes historical analysis (Tang 2019), parallel comparison of multiple translations (Amarantidou 2023), and the conceptual transformation of key terms such as Dao 道 (way or sense), De 德 (virtue or morality), and *xiang* 象 (symbolic imagery, images) (Zhu and Song 2022). However, the investigation of translation strategies from a "paratext" perspective remains relatively scarce.

The term "paratext" was first propounded by the French literary scholar Gérard Genette. It delineates the array of linguistic and non-linguistic forms accompanying a work, which, while not constituting the main text, invariably surround, extend, and present it (Genette 1997, p. 1). In translation works, paratexts generally include the book cover, title page, preface, footnotes, annexes, illustrative material, and the like. Paratexts mediate and enrich the dynamic nexus between the book and its readership, thus "facilitating the acceptance and consumption of the book" (Geng 2016, p. 105). Recently, scholars have used the concept of "paratext" to look at Daoist materials. For instance, by analyzing the paratexts in Daoist texts, Elena Valussi has delineated the evolution of "female alchemy" (女丹 *nüdan*) (Valussi 2008). Similarly, through meticulous investigation on the abundant talismans and diagrams in Daoist scriptures, Dominic Steavu has revealed that such paratexts

are not merely key tools for reading and understanding the texts, but also central elements in the practice and experiential aspects of religious rituals (Steavu 2019). Such research effectively demonstrates the active role of paratexts in the construction of textual meaning.

The *Daodejing* is a classic filled with cultural connotations but presented in an exceptionally simple stylistic modality. In its translation across cultures, it inevitably experiences interpretative deviations, elisions, and metamorphoses. Owing to the constraints of the source text, the translated text frequently lacks adequate interpretive space to faithfully reproduce the abundant cultural intricacies inherent to the original composition. Therefore, regardless of the language in which the *Daodejing* is translated, it is generally accompanied by multifaceted and content-rich paratexts. These paratexts not only reflect the interplay between the translator's personal interpretation and the original text's inherent meaning, but also emerge as an important avenue for the reconstruction of the *Daodejing* within foreign cultural terrains. Considering these factors, this study endeavors to analyze Richard Wilhelm's translation—lauded by Hermann Hesse as the "optimum translation" (Hesse 1921, p. 250) in the German world—to discern the role of paratexts in the translation and reception of the *Daodejing*.

Since its publication, Wilhelm's translation has influenced European intellectuals like Hermann Hesse, Carl Gustav Jung, and Bertolt Brecht, and has become an indispensable reference for numerous subsequent translators. Until 2022, Wilhelm's translation has been reprinted nearly 30 times by diverse publishing houses, solidifying its stature as the most influential German translation of the *Daodejing*. By examining the function of paratexts, this study helps elucidate the elements contributing to the success of Wilhelm's translation. This perspective contributes to the research of transcultural adaptation models of the *Daodejing* and the interactive relationship between Chinese literature and world literature.

## 2. The Paratexts in Wilhelm's Translation

Wilhelm's translation has two distinct editions. The first edition was published in 1911, designated as the second volume within the series entitled "Chinese Religion and Philosophy". After completing the initial version, Wilhelm spent several years meticulously re-examining his translation and removing sentences that were clearly imbued with theological elements (Tan 2011). The second edition was completed in 1916, yet its publication was deferred until 1957 under the editorial stewardship of his spouse Salome Wilhelm. Considering the milestone significance of Wilhelm's 1911 edition within Laozegetics (老學 *Laoxue*)[1] in Germany, this paper selects this edition as its focal point. The investigation predominantly gives attention to the paratexts in this translation, encompassing book cover, title pages, illustrations, foreword, introduction, footnotes, post-textual interpretation, and appendix.

### 2.1. Cover, Title Pages, and Illustrations: Identifying Book Categories and Attracting Readers

The cover operates as a gateway ushering readers into a book. Prior to readers' immersion into the main text, the cover has already begun to transfer information. As posited by Kratz, "Besides telling the kind of book, covers are also intended to convey the identity of each book, to attract and pique the buyer/reader's interest" (Kratz 1994, p. 186). Wilhelm's translation is enrobed in a hardcover, displaying its elegance and solemnity, which augments the tactile and aesthetic qualities of this book (Figure 1). The cover is adorned with a warm, orange-yellow background with the German title "*LAOTSE VOM SINN UND LEBEN*" (*Laozi on SENSE and LIFE*) at its center. Following Genette's identified basic elements of a cover—title, author (translator), and publisher—this cover solely presents the title in an extremely minimalistic fashion, indicative of a realistic cover design commonly seen in serious works (Haslam 2006, p. 165). Through this cover, readers can preliminarily perceive the stylistic tenor of this classic. Wilhelm's translation maintains the consistent cover design of the "Chinese Religion and Philosophy" series, not only unifying the coherent visual narrative of the collection but also promoting brand recognition, which in turn enhanced the cumulative market traction of the series.

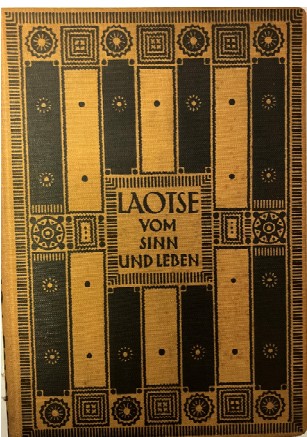

**Figure 1.** The book cover.

The title pages of Wilhelm's translation also exhibit thoughtful deliberation. The first page displays the distinguishable lion emblem of Diederichs publisher.[2] The second page integrates Chinese copyright details, overlaid with a semi-transparent rice paper on the upper portion (Figure 2). On this rice paper, phonetic annotations and German words correspond directly to the woodcut Chinese characters beneath, offering readers information about the title, translator, and the publisher. On the third page, a graphic representation of Laozi is presented, characterized by his graying hair and frost-like temples. He is illustrated holding a stone tablet bearing inscriptions, embodying the image of a wise sage. The fourth page presents German copyright specifics, explicitly stating that Wilhelm's translation is "translated and elucidated from Chinese", thereby setting it apart from other German translations originating from European languages and accentuating the authority and credibility of Wilhelm's work.

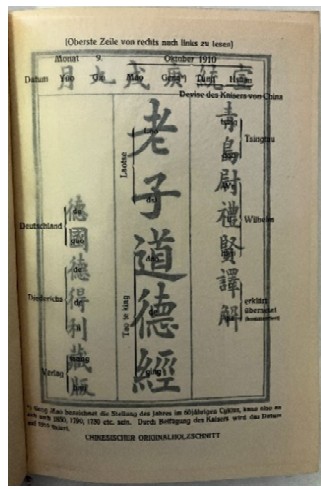

**Figure 2.** Chinese copyright.

On pages five and six, a highly distinctive, traditional, and realistic Chinese-style painting is showcased (Figure 3). This painting employs the "pointillism technique" (點苔 *diantai*) to delineate the textures of trees and stones, creating spatial layers in the forested landscape. It integrates the "ancient gossamer drawing style" (高古遊絲描 *gaogu yousi miao*) to manifest the detailed and natural facial features of the characters, while their clothing is elegant and flowing. The painting incorporates plots such as a border officer bidding adieu to Laozi, a servant carrying books, and Laozi riding a cow while looking back, which vividly depicts the legend of "Laozi Going Through the Pass" (老子出關 *laozi chuguan*). It is worth noting that, the title pages of a book typically consist of 1–2 pages, which means the six title pages in Wilhelm's translation are notably atypical.

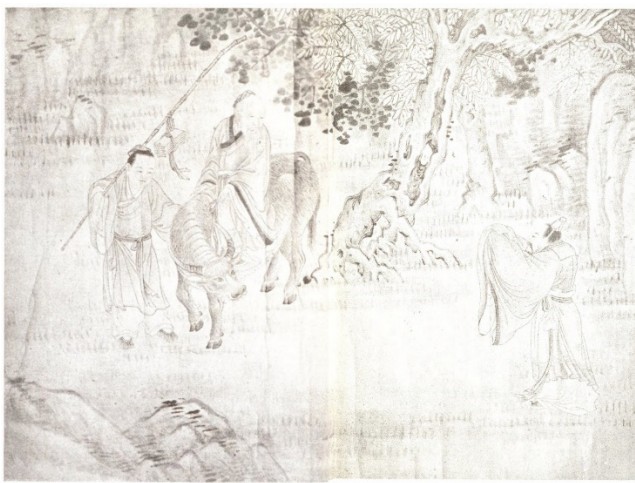

**Figure 3.** Laozi Going Through the Pass.

In fact, over the 40 years since the full translation of the *Daodejing* appeared in 1870, none of the existing eight translations[3] have featured as many title pages and illustrations as Wilhelm's version. Both Wilhelm and Diederichs attached importance to the book's illustration. In a letter discussing the publication of the *Daodejing*, Wilhelm wrote: "There is a conventional depiction of him (Laozi) riding an ox, which is circulating in countless good and bad reproductions. I would like to secure an artistically valuable presentation for this purpose" (Wilhelm 1967, p. 178). He and the publisher aimed to meet "evolving audience needs and the precise positioning of cultural dissemination" (Zhang 2020, p. 82), using the title pages and illustrations to evoke an initial resonance of an oriental flair and Chinese atmosphere. This strategy catered to general readership, reflecting Wilhelm's focus on popularization and marketability. In 1938, the renowned German poet Bertolt Brecht was inspired by this illustration to create his poem *Legend of the Origin of the Daodejing on Laozi's Path of Exile* (*Legende von der Entstehung des Buches Taoteking auf dem Weg des Laotse in die Emigration*) (Tan 2012, p. 121), which undoubtedly demonstrated Wilhelm's success in the selection of paratexts.

*2.2. Foreword and Introduction: Background Supplement and Text Overview*

In a translated work, the foreword and introduction typically delineate the translator's translation strategy, augment background knowledge of the original text, and summarize its themes. Its chief function is "to ensure that the text is read proper" (Genette 1997, p. 197). The prefatorial segment to Wilhelm's translation is composed of three distinct components: (1) the foreword, penned by Wilhelm himself; (2) the introduction, also authored by Wilhelm; (3) another introduction entitled "*Laozi's Reflections on Human Society*", contributed by Dr. Harald Gutherz, an academic affiliated with the Department of Jurisprudence at Qingdao Dehua University. It is necessary to emphasize that as Wilhelm transitioned from a missionary to a translator, he did not intend to create scholarly translations solely for the German sinological community. Instead, he regarded the general German public as his primary readership. As early as his initial attempt to translate the *Analects* (論語 *Lunyu*) in 1904, Wilhelm recognized that existing German translations of Chinese classics were often arcane and perplexing, largely because these translations were excessively rigid (Wilhelm 1904, p. 34). Consequently, whether in handling the translation or the paratexts, Wilhelm attached importance to their accessibility and acceptability.

In the foreword, Wilhelm primarily addressed two pivotal questions: why did he undertake the translation of the *Daodejing*? And how should this mysterious and difficult classic be translated and interpreted? Before Wilhelm's translation appeared, there were at least eight distinct German translations of the *Daodejing*. Notably, the version by the theologian Victor von Strauss stands out, having been frequently cited by renowned sociologists like Max Weber. The *Daodejing* was one of the most scrutinized Chinese classics

within the German-speaking cultural sphere at that time. In Wilhelm's own words: "If one attempts to translate *Laozi* in this day, it requires an explicit apology in front of all specialized Sinologists." (Wilhelm 1911, p. I). However, Wilhelm also candidly pointed out that many scholars did not diligently study the original Chinese text, but rather transliterated the *Daodejing* into German from English or French versions. Thus, in his translation, he emphasized returning to the original source, relying on Chinese documents as the primary references, with European texts serving merely as supplementary material citation. He strived to faithfully reproduce the Chinese classic, allowing Laozi "to express his own voice once more" (Wilhelm 1911, p. II).

The introduction, spanning 29 pages and totaling 10,977 words, stands as the most critical component of the paratexts in Wilhelm's translation. Initially, Wilhelm enumerated records of Laozi from Chinese classics such as *Shiji* 史記 (Records of the Grand historian), *Liji* 禮記 (the Classic of Rites), *Kongzi Jiayu* 孔子家語 (the Family Sayings of Confucius), and the *Analects*. Selecting the narrative of "Laozi Going Through the Pass" from the *Shiji*, Wilhelm recreated the enigmatic legend behind the creation of the *Daodejing*. Subsequently, Wilhelm discussed the chronology of the *Daodejing*'s composition by citing ancient classics like the *Analects*, *Liezi Chongxujing* 列子沖虛經 (*Liezi*: Classic of Simplicity and Vacuity, hereafter abbreviated *Liezi*), *Zhuangzi Nanhuajing* 莊子南華經 (*Zhuangzi*: Classic of Southern Flower Country, hereafter abbreviated *Zhuangzi*), and *Huainanzi* 淮南子 (the Book of *Huainanzi*). The veneration of the *Daodejing* by emperors of the Han dynasty (漢代, 202BC–220AD), as well as the abundant commentaries and interpretations across successive dynasties, underscores its eminent position in ancient China. Wilhelm then referenced the Japanese scholar Taizai Chuntai 太宰春台 (1680–1747) to contrast the philosophical divergences between Confucius and Laozi, further contextualizing the historical backdrop of the *Daodejing*.

Wilhelm posited that Laozi's thought harbor an enduring value that transcends time, "Laozi are increasingly starting to be picked up in Europe nowadays" (Wilhelm 1911, p. XIII). At the end of the 19th century and the beginning of the 20th century, Germany, amidst rapid industrialization and modernization, began to experience a modernity crisis of nihilism (Xu 2023, p. 62). This spiritual dilemma led many Germans to seek relief and guidance in the inner peace offered by Daoism. As Carl Gustav Jung expressed: "The spirit of the East is truly at the gates. Therefore, it seems to me that the realization of meaning, the seeking of Dao, has already become a more collective phenomenon among us to a much greater extent than is generally thought" (Jung 1982, p. XVIII). Against the backdrop of Nietzsche's prevailing philosophy and the waning of Christian faith, the Daoism to some extent took on the role of a new type of "salvation religion" for many Germans (Detering 2008, p. 27). However, Wilhelm opposed the notion of viewing Daoism thought merely as a religious concept. He specifically cautioned readers: "What is commonly referred to as Daoist nowadays can be traced back to the animistic folk religion of ancient China" (Wilhelm 1911, p. XIV). Although the *Daodejing* is a foundational text of Daoist, it should not be solely regarded as a religious text. In reality, it comprises Laozi's exploration of the world's origins and a system of metaphysical philosophical thought.

What follows are two essays elucidating the core ideas of the *Daodejing* from metaphysical and sociological perspectives. The former is authored by Wilhelm while the latter by Gutherz. Interestingly, despite approaching the *Daodejing* from disparate angles and without prior collaboration, both essays manifest many similar viewpoints. Below Gutherz's contribution, the editor appended a footnote: "In the following essay, the reader will encounter trains of thought at two points that are familiar to him from the previous work. Nevertheless, the essay was printed unchanged because it was written without knowledge of the preceding work and because the apparent repetition might be of interest as the same result found through different paths." (Wilhelm 1911, p. XXVI).

Broadly speaking, the foreword and introduction employ multifaceted narratives, encompassing literature review, background introduction, textual commentary, and intellectual critique. From these narratives, an optimal mode of textual interpretation can be

deduced, ensuring the translated text balances readability, acceptability, scholarly rigor, and fidelity.

*2.3. Footnotes and Post-Textual Interpretation: Knowledge Integration and Meaning Clarification*

Footnotes, endnotes, and post-textual interpretation encompass definitions of specialized terms, clarifications of facts, amplifications of viewpoints, and cited sources. Such paratexts help eliminate potential ambiguities, enhance the clarity of the text, and ensure that readers can better understand the author's viewpoints. In most translation works, these functions are conventionally undertaken by footnotes or endnotes (Genette 1997, p. 327). Contrary to convention, Wilhelm demonstrated a marked reluctance to employ footnotes and endnotes in this translation. But the post-textual interpretation extends to approximately 28 pages, comprising close to 9500 words. This is unusual, especially when compared to other German translations. For instance, Strauss's translation employs both prose and rhymed poetry, and is annotated extensively via lengthy footnotes, extending to 440 pages. Plaenckner's version, translated in prose, lacks annotations but appends extensive interpretation after each chapter, totaling 455 pages. Wilhelm's translation is entirely expressed in rhymed poetry, with only a single footnote, and interpretation appear after the end of all main texts, resulting in a total of 159 pages. Upon comparison, it is evident that the length of Wilhelm's translation is moderate and the main text is not interrupted by long annotations or interpretations, which not only ensures fluidity for general readers but also retains academic value. This very aspect was lauded by Hesse, who praised Wilhelm's version as "more potent, lucid, possessing a distinct personal touch, and hence, more comprehensible" (Hesse 1911, p. 33).

Furthermore, a closer examination of Wilhelm's other translation works reveals that the use of paratexts varied according to the genre of each text. The following table provides statistics on some of the paratexts in the "Chinese Religion and Philosophy" series (Table 1). For the *Analects* and *Mengzi* 孟子 (the Book of *Mengzi*), which primarily consist of sayings and dialogues, Wilhelm utilized footnotes to timely supplement historical knowledge and cultural context, thereby preventing the loose dialogues from creating reading barriers. In contrast, for *Zhuangzi*, *Liji* and *Lüshi Chunqiu* 呂氏春秋 (Spring and Autumn of Lübuwei), which are written in essay format and have relatively concise themes in each chapter, endnotes were used to avoid interrupting the readers. Additionally, introduction is prefaced before each chapter of *Zhuangzi* and *Liji*, serving a similar role in aiding comprehension. Facing the *Daodejing* and *Yijing* 易經 (the Book of Changes), which offer vast interpretative space, Wilhelm preferred using post-textual interpretation. Unlike footnotes and endnotes, such paratexts often contain stronger expressions of the translator's personal perspective while supplementing information.

**Table 1.** Paratexts for the "Chinese Religion and Philosophy" Book Series.

| Chinese Classics | Foreword (Page) | Introduction (Page) | Footnote (Item) | Endnote (Item) | Post-Textual Interpretation (Page) |
|---|---|---|---|---|---|
| *Lunyu* 论语 | 3 | 31 | 407 | - | - |
| *Daodejing* 道德经 | 3 | 29 | 1 | - | 25 |
| *Liezi* 列子 | 2 | 21 | - | - | 42 |
| *Zhuangzi* 莊子 | 2 | 17 | - | 463 | - |
| *Mengzi* 孟子 | 1 | 18 | 525 | - | - |
| *Yijing* 易經 | 2 | 11 | 50 | - | 70 |
| *Lüshi Chunqiu* 呂氏春秋 | - | 13 | 7 | 731 | - |
| *Liji* 禮記 | - | 18 | - | 588 | - |

In the footnote and post-textual interpretation, Wilhelm interpreted 18 terms such as *taiji* 太極 (supreme ultimate), *wuji* 無極 (ultimate of Nothingness), *chugou* 芻狗 (straw dogs), *shengqi* 神器 (mental things), *jisi* 祭祀 (sacrificial rite), and *shengren* 聖人 (sage of wisdom), which are steeped in rich cultural traditions and societal contexts. The aim is to approximate the original context as closely as possible, thereby reducing potential reading barriers for the audience. The sole footnote appears in Chapter 5, which reads "Not love in the manner of men has Nature: To her, the creatures are like straw dogs. Not love in the manner of men has the sage: To him, his people are like straw dogs" (天地不仁, 以萬物爲芻狗; 聖人不仁, 以百姓爲芻狗) (Wilhelm 1911, p. 7). In this instance, Wilhelm referenced *Laozi zhu*老子注 (Commentary on *Laozi*) to elucidate straw dogs:

> In sacrificial rites, dogs were made of straw, which were festively decorated during the sacrifice, but once they had served their purpose, were carelessly discarded. (Wilhelm 1911, p. 7)

In the original text of the *Daodejing*, the connotation of straw dogs is negative, which greatly deviates from the dog's image in German native culture, symbolizing loyalty, kindness, and wisdom. This difference might be confusing for readers, prompting Wilhelm to employ a concise footnote for clarification. However, other terms are unique to Chinese culture or do not have entirely opposites in German culture. As such, it is not necessary to interrupt readers' fluent reading experience with footnotes. Instead, more detailed interpretations are furnished at the end of the text. For example, when clarifying *taiji* and *wuji*, Wilhelm also added two illustrations as support:

> To elucidate this unity, Laozi refers to the symbolic figure of *taiji* (often translated as "Primordial Beginning"), which has significant resonance in ancient Chinese thought and has been particularly used in endless variations and adaptations, representing the intertwining of the positive and negative (Figure 4).

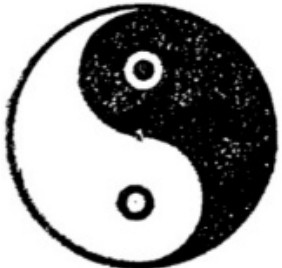

**Figure 4.** *Taiji* 太極.

> Wherein the white half of the circle, containing within itself a black circle with a white dot, signifies the positive, masculine, and luminous principle. In contrast, the correspondingly designed black half symbolizes the negative, feminine, and dark principle. This symbolic figure is likely alluding to the profound mystery of the unity between the existent and the non-existent (=μη ὸν, as consistently referred to by Laozi whenever discussing the "non-existent"). An even deeper mystery within this enigma would be the so-called *wuji* (translated as "Non-Beginning", even beyond *taiji*), representing a chaotic state before any distinctions are made, typically represented by a simple circle (Figure 5). It can be described as the pure possibility of existence, akin to chaos. (Wilhelm 1911, pp. 89–90)

Considering that Laozi has used *wuji* to describe the dialectical relationship between existence and non-existence, Wilhelm similarly employed these concepts to expound some chapters of the *Daodejing*. The above paratext is used to explain the sentence in Chapter 1, "Beyond the nameable is the origin of the world. This side of the nameable is the birth of creatures" (無名, 天地之始; 有名, 萬物之母) (Wilhelm 1911, p. 3). To Wilhelm's understanding, "beyond the nameable" closely relates to *wuji*, while the notion of "this side of

the nameable" resonates with *taiji*. When translating the sentence from Chapter 28, "Thus, he is not without eternal LIFE (LEBEN), and he can return to the Unborn (*wuji*)" (常德不忒, 複歸于無極) (Wilhelm 1911, p. 30), Wilhelm expressed *wuji* as "Unborn" and explained that "Unborn is the chaotic state of intertwined substances before the inception of *taiji*, that is, before the commencement of creation" (Wilhelm 1911, p. 100). In elucidating the sentence from Chapter 42, "The SENSE (SINN) generates the Unity. The Unity generates the Duality. The Duality generates the Ternarity. The Ternarity generates all creatures" (道生壹, 壹生二, 二生三, 三生萬物) (Wilhelm 1911, p. 47), Wilhelm explained:

> The Unity refers to *wuji*, the Duality refers to *taiji*—with its division into *yang* 陽 and *yin* 陰,[4] and the Ternarity signifies "the infinite vitality", namely, the spirit, is, so to speak, the medium of the unification of the two dual forces. (Wilhelm 1911, p. 103)

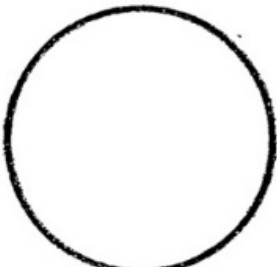

**Figure 5.** *Wuji* 無極.

Indeed, this corollary appears even earlier in the introduction:

> The Duality encompasses the opposites of light and darkness, of male and female, of positive and negative, and the Ternarity emerging from *taiji* represents the process wherein opposing entities combine, counteracting each other, subsequently engendering myriad entities. (Wilhelm 1911, p. XXIII)

Through repeated elucidation and emphasis of these concepts, Wilhelm facilitated readers in acquiring a more profound comprehension of the dialectical unity central to the *Daodejing*.

Certainly, the ability to implement these repetitions owes to Wilhelm's extensive use of intertextual references, as shown in Figure 6. For instance, in the introduction, when addressing *taiji* and *wuji*, Wilhelm reminded readers to consult the interpretation of Chapter 42. And in the interpretations of Chapters 28 and 42, he also guides readers to refer to the interpretation of Chapter 1. Such internal intertextual pointers amount to a total of 71 instances, reaching a level of meticulous detail. This serves to remind readers of the inner connections of the text when necessary, helping them locate related content and thus promoting a holistic understanding of the translation

Moreover, Wilhelm frequently cites other works to provide readers with opportunities for comparative analysis or extended reading. Statistics show that the *Analects* and the *Bible* are the most frequently referenced works. Other cited texts include Chinese classics like *Liezi*, *Shanhaijing* 山海經 (the Classic of Mountain and Sea), and *Yijing*, with translations by predecessors like Strauss and Carus. Western literary works like Goethe's *Faust* and Schiller's *An die Freude* (*Ode to Joy*), philosophical propositions by Spinoza, Nietzsche, and Heraclitus, academic treatises by Otto Frank and Chavannes, and even Latin proverbs are also referenced.

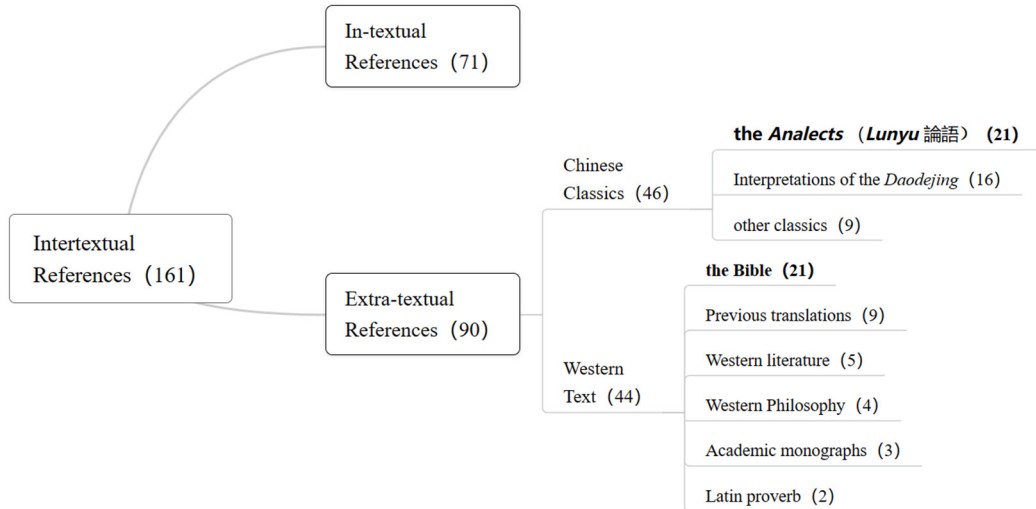

**Figure 6.** Statistics on the intertextual references.

*2.4. Appendix: Academic Supplements and Book Series Promotion*

The appendix of Wilhelm's translation includes bibliographic references and a list of the "Chinese Religion and Philosophy" book series. In the foreword, Wilhelm articulated that his main reliance would be on a myriad of Chinese original literatures, relegating European sources to supplementary material citation. Accordingly, in the bibliography, he listed Chinese commentaries of the *Daodejing*, alongside representative translations in English, French, and German. *Heshanggong zhangju* 河上公章句 (Heshanggong's Commentary on the *Laozi*) and Wang Bi's 王弼 (226–249) *Laozi zhu* 老子注 (Commentary on *Laozi*) serve as the primary references for Wilhelm's translation. Additionally, other references include Lu Deming's 陸德明 (550–630) commentary *Laozi yinyi* 老子音義 (Pronunciation and Meaning of the *Laozi*), Xue Hui's 薛蕙 (1489–1541) commentary *Laozi jijie* 老子集解 (Collection and Annotations of the *Laozi*), Hong Yingshao's 洪應紹 (around 1612) commentary *Daodejing ce* 道德經測 (Interpretation of the *Daodejing*), Wang Fuzhi's 王夫之 (1619–1692) discourse on *Laozi*, and the Japanese scholar Taizai Chuntai's *Laozi tejie* 老子特解 (Special Interpretation of the *Laozi*). These texts were compiled during the Han 漢 (202BC–220AD), Tan 唐 (618–907), Ming 明 (1368–1644), and Qing 清 (1636–1912) dynasties, spanning a period of several millennia. Among its foreign translations, the English version by James Legge and the German version by Victor von Strauss are esteemed works. These references not only help readers delve deeper into the *Daodejing*, but also enhance the academic value of Wilhelm's translation.

The back cover of this book features a list from the "Chinese Religion and Philosophy" series, showing the distribution volume of published books and apprising upcoming publications. As Diederichs aptly posited, "one can promote the old books with every new release" (Diederichs 1936, p. 410). The publishing list on each volume's endpaper aims to bolster the overall impact of the series, facilitating the circulation of Chinese classics in the German-speaking world.

## 3. The Constructive Role of Paratexts in Wilhelm's Translation

Wilhelm translated the *Daodejing* into rhymed poetry, preserving the original text's cultural imagery, rhetorical characteristics, and compositional style (Hua 2012, p. 125). Concurrently, his translation includes diverse paratexts, ensuring its literary value and sinological significance. Upon deeper analysis, it is evident that through these paratexts, Wilhelm constructs a philosophical framework of Daoism, compares Confucianism and Daoism thought, and broadens the dialogue between Chinese philosophical thought and Western intellectual traditions, thereby bestowing upon the *Daodejing* a renewed vitality in the German-speaking world.

### 3.1. Constructing a Philosophical Framework of Daoism

In the introduction, Wilhelm clarified that "The entire metaphysics of the *Daodejing* is built upon a fundamental intuition that is inaccessible to strict conceptual fixation, and to have a name, Laozi 'provisionally' designates it with the word Dao" (Wilhelm 1911, p. XV). When translating Dao, Wilhelm avoided existing translations such as "God", "Way", "intellect", "word", and "λόγος" (logos). He believed that: "Fundamentally, the expression holds minimal significance, as for Laozi himself, it represents merely an algebraic symbol for an ineffable concept. The preference for a German term in a German translation primarily stems from aesthetic considerations" (Wilhelm 1911, p. XV). Therefore, Wilhelm, deeply knowledgeable in biblical linguistics and familiar with German literary masterpieces, selected "SINN" (sense, meaning), a term utilized by Goethe in *Faust*, which holds an equally enigmatic and inscrutable connotation within the German context.

Recognizing the linguistic non-equivalence between different languages, Wilhelm applied a hermeneutical approach. He first acknowledged the existence of deviations, and then compared the diverse meanings of terms. He explored the semantic meanings of the Chinese term Dao, encompassing "direction", "truth", "talk", "guidance", and "pathway". Then, he examined the different semantic dimensions of the German term "SINN", such as "pathway", "direction", "intrinsic inclination", and "sense" (Wilhelm 1911, p. XV). Through this detailed comparison, Wilhelm suggested a close concordance between Dao and "SINN".

Likewise, his interpretation of De diverges from traditional translational norms. Historically, De has often been translated as "virtue". However, Wilhelm, well-versed in ancient Chinese philosophical texts, draws connections to a sentence from *Zhuangzi*: "What sustains beings to come into existence is called De" (物得以生, 謂之德). This narrative is congruent with, and perhaps even extends the perspective articulated in Chapter 51 of the *Daodejing*: "The SENSE (SINN) generates. LIFE (LEBEN) nourishes. The being shapes. The power completes" (道生之, 德畜之, 物形之, 勢成之) (Wilhelm 1911, p. 56). Wilhelm thus posited that the essence of De closely parallels the thought in the *Gospel of John*: "In him was life, and that life was the light of men" (Wilhelm 1911, p. XVI). In light of this narration, Wilhelm audaciously translated De as "LEBEN" (life).

To emphasize the specific connotations of "SINN" and "LEBEN", Wilhelm deliberately rendered these terms in uppercase. This stylistic choice draws inspiration from representation of "GOTT/HERR" (God) in the German *Bible*. For German readers familiar with Christian tradition, the uppercase "SINN" and "LEBEN" readily evoke associations with the German *Bible* as translated by Martin Luther. The unique cultural encoding behind these two core concepts not only places the *Daodejing* in an overlay region beneath German literature and theology but also evokes recollections of the *Bible* and *Faust* among German readers. Utilizing this approach, Wilhelm intimated that this Chinese classic holds a distinguished position, by no means inferior when compared to the *Bible* and *Faust*, thereby considerably heightening readers' appreciation of the *Daodejing*.

After explaining the two most fundamental concepts, Wilhelm begun to explore the standpoint from which Laozi constructs his metaphysics. Unlike the ancient Greek philosophical systems, which seek the essence and universal principles from the external world, Chinese philosophical systems, whether of Laozi or Confucius, primarily concentrate on the realm of the human spirit. Based on the statements in Chapters 12, 38, and 72, Wilhelm summarized Laozi's contemplation as follows:

> Every principle taken from external experience will be refuted and become obsolete over time, because as human progress advances, so does the understanding of the world. On the contrary, what is recognized from central experience (from the inner light, as expressed by the mystics), remains irrefutable, provided it was otherwise purely and correctly perceived. (Wilhelm 1911, p. XVII)

Of course, Laozi is not considering singular, accidental mental experiences but the "pure self" inherent to the human group beyond the individual. The natural essence of hu-

manity forms a continuous, cyclical unity, conforming to nature and being fundamentally consistent with all things, which is De. And if one follows nature even further, beyond humanity, Dao will be found. That is, just as Dao is in humans, De exists in the world purely as spontaneity.

After elucidating Laozi's principles for explaining the world, Wilhelm continued to explore how Laozi has deduced the practical path from these highest principles. Wilhelm believed that this is precisely where Laozi struggles, not only in his personal relationship to the external world, as discussed in Chapter 20, but also in deriving the external world from Dao (Wilhelm 1911, p. XXII). Nevertheless, Laozi strives to indicate the possible direction of Dao toward reality, namely, "The SENSE (SINN) generates the Unity. The Unity generates the Duality. The Duality generates the Ternarity. The Ternarity generates all creatures" (道生壹, 壹生二, 二生三, 三生萬物) (Wilhelm 1911, p. 47). As discussed above, the Unity gives rise to contradictory opposites, and the Duality develops in contradiction and opposition. For this reason, Laozi proposes the doctrine of *wuwei* 無爲 (non-action), which does not require individuals to waste life passively but states that "action" would disrupt the balanced opposing forces. By following the natural laws and the primal origin of all things, one can maintain harmony and stability in the universe. By adhering to Dao and practicing *wuwei*, individuals can also live in balance and harmony.

### 3.2. Comparing the Thought of Confucianism and Daoism

Existing studies have predominantly focused on Wilhelm's extensive citation of Western literature, especially the texts of the *Bible*. As Xu suggests, "This approach allows readers to find familiarity within unfamiliar texts, thereby constructing a bridge between Chinese philosophical thought and Western intellectual traditions where mutual invention and comprehension in cultural spirit are possible" (Xu 2023, p. 65). However, as shown in Figure 6, the frequency of Wilhelm's references to Chinese literature is comparable to that of Western literature, with the number of citations from the *Analects* coincidentally aligning with those from the *Bible*. Before building a bridge for Sino-Western cultural communication, Wilhelm had already created internal communication between Confucianism and Daoism.

Wilhelm's comparison of Confucianism and Daoism is multifaceted. In the introduction, he begun by analyzing the character traits of the representative figures of these two schools, Confucius and Laozi. Wilhelm posited that Confucius was actively engaged with society, and hence, the *Analects* is filled with evaluations of significant figures from his time and from history. In contrast, Laozi was reluctant to engage with society and the *Daodejing* almost never evaluates any historical events or figures. However, Wilhelm also argued that this is precisely the reason why "Laozi can transcend time and space to exerts such great effects in Europe"(Wilhelm 1911, p. VIII). In a letter written to Eugen Diederichs in 1910, Wilhelm mentioned that he believed the *Daodejing* would be more popular than the *Analects* because understanding the *Daodejing* "requires far fewer historical prerequisites" (Wilhelm 1967, p. 178).

Confucius was actively engaged with society, traveling tirelessly among various states in an endeavor to find monarchs willing to heed his teachings and, thus, to rescue the populace from turmoil and calamity. In contrast, Laozi believed that "the disease afflicting a state cannot be cured by a particular remedy", asserting that "any form of intervention would only exacerbate chaos. It is preferable to allow the afflicted 'body' to rest and await natural forces (Dao) to effect its restoration" (Wilhelm 1911, p. IX). Whether it is Confucius's societal engagement and salvific ideology or Laozi's naturalistic and non-interventionist approach, both are continuations of ancient Chinese spiritual traditions. Moreover, Confucius did not oppose Laozi's principle of "non-action", but rather regarded it as the highest insight in Chapter 15 of the *Analects* (Wilhelm 1911, p. 90).

Dao is not only at the core of Daoism but also central to Confucianism, although its interpretation is different between the two. In Confucianism, Dao refers to the way for a monarch to govern the state, while in Daoism, Dao is the primal origin of all things.

Additionally, Wilhelm believed that there are notable differences between Confucianism and Daoism in their treatment of *li* 禮 (the ritual system of the Zhou Dynasty). Confucius placed *li* at the core of his philosophy, believing that restoring the rites of Zhou 周 (1046 BC–256 BC) dynasty could bring an end to social unrest. However, Laozi viewed *li* as a sign of social regression. He advocated for a more ancient value system, following thought from before the Zhou Dynasty (Wilhelm 1911, p. X).

Beyond the systematic comparisons made in the introduction, Wilhelm frequently referenced the *Analects* in his post-textual interpretation. For instance, when elucidating the phrase "Respond to resentment with LIFE (LEBEN)" (報怨以德) (Wilhelm 1911, p. 68) in Chapter 63, Wilhelm expressed:

> The sentence: "Respond to resentment with LIFE (LEBEN)" usually translated as: "Repay wrong with kindness", plays a certain role in the discussions of the time. Laozi justifies it in Chapter 49 by stating that our actions necessarily arise from our nature, thus we can only be good. He thus surpasses the concept of "reciprocity", which occupies such an important place in post-Confucian systems. Confucius had doubts about this concept for reasons of state justice (see his statement on the question in the *Analects*, book XIV. 36, page 163), although he has acknowledged the principle for individual morality (see *Liji*). (Wilhelm 1911, p. 109)

In the *Analects*, De encompasses meanings such as "nature", "essence", "spirit", and "force", which closely align with the connotations of De in the *Daodejing*. However, considering that "Respond to resentment with LIFE (LEBEN)" is often interpreted by later generations as repaying resentment with kindness, Wilhelm found it necessary to clarify this view. Thus, he compared similar viewpoints in the *Analects* and explicated the implications of this sentence at both the individual and state levels.

Cross-textual references to Chinese classics like this are numerous in Wilhelm's translation. However, references to the *Analects* in the *Daodejing* all come from his translation published in 1910, which is meticulously accompanied by chapter locations and page numbers for the timely consultation by readers. This not only enables German readers to intuitively understand the differences between Chinese Confucianism and Daoism, but also helps to boost the sales of previous books with the publication of new ones. Wilhelm continued this method of cross-edition referencing in his subsequent translations. For example, in *Liezi* and *Zhuangzi*, Wilhelm tirelessly cites translations of the *Daodejing*. This not only maintains the continuity of the series of books but also systematically presents Chinese philosophical thought as a coherent whole.

### 3.3. Broadening the Dialogue between Chinese Philosophical Thought and Western Intellectual Traditions

Prior to Wilhelm's translation, most translators interpreted the *Daodejing* predominantly as a religious theological text, with Strauss's version being a typical example. Such interpretive norms had already been deeply entrenched among German readers, making developing a new interpretation no easy task. Therefore, Wilhelm chose to follow the path of his predecessors, extensively referencing the *Bible* to help readers understand the *Daodejing*. However, unlike Strauss, who attempted to demonstrate that Christian thought had already existed in ancient China, Wilhelm cited the *Bible* for the purposes of comparison. He consistently regarded the *Daodejing* as an indigenous Chinese philosophical system, rather than merely a religious text.

For instance, when elucidating the passage from Chapter 14, "One looks for him and does not see him: His name is: The Equal (夷 *yi*). One listens for him and does not hear him: His name is: The Subtle (希 *xi*). One reaches for him and does not grasp him: His name is: The Minute (微 *wei*)" (視之不見, 名曰夷; 聽之不聞, 名曰希; 搏之不得, 名曰微) (Wilhelm 1911, p. 16), Wilhelm wrote:

> The three names of the SENSE (SINN): "The Equal" "The Subtle" "The Minute" signify its supernatural qualities. Attempts to read the Hebrew name of God

from the Chinese sounds I, Hi, We may now be at an end. (Victor von Strauss, as is well known, still believed in this; see his translation.)

The fact that the view of SENSE (the deity) outlined here has some parallels in Israelite teachings is not to be denied; see especially the passages in Chapter 33 of the *Exodus* and Chapter 19 of the *Book of Kings* for our section. However, such agreements are understandable enough even without direct contact. This view of the deity simply represents a certain stage of development of human consciousness in its understanding of the Divine. Moreover, the fundamental difference between Laozi's impersonal pantheistic conception and the sharply defined historical personality of the Israelite God must not be overlooked. (Wilhelm 1911, p. 94)

Strauss once deciphered the Hebrew name Jehovah from the syllables of the Chinese word "Yi-Hi-Wei", positing that Chinese people had early acknowledgment of the existence of Jehovah. He suggested that during the destruction of ancient Israel and Judah (in 720 BCE and 586 BCE, respectively), Jewish refugees might have fled to China. This could have provided Laozi the opportunity to learn the name of God from them. He then proposed that Laozi subtly incorporated and concealed the three characters of "Jehovah" within Chapter 14 of the *Daodejing* (Strauss 1870, pp. 61–75). Wilhelm explicitly opposed such a far-fetched conclusion, considering this recognition merely a manifestation of collective consciousness appearing at a certain stage of human development, rather than a result of one side influencing the other. He thus delineated the distinct differences between Laozi's thought and Christian doctrines.

In addition to establishing a religious comparative pathway distinct from his predecessors, Wilhelm also incorporated texts from Western classical philosophy and German literature into his translation. As mentioned above, when discussing Laozi's metaphysics, Wilhelm compared it to ancient Greek metaphysical thought. When interpreting the sentence "The Ternarity generates all creatures" (三生萬物) (Wilhelm 1911, p. 47), Wilhelm reminded readers, "It deserves noting how the rational philosophy in Laozi treads precisely the same paths as it does in Hegel, two and a half millennia later" (Wilhelm 1911, p. XXIII). This departs from the previous religious interpretation that likened "The Ternarity generates all creatures" to the Holy Trinity. Further, when discussing reducing human intervention and advocating *wuwei*, Wilhelm even cited a sentence from *Faust*, "Reason becomes nonsense, benefit turns into torment" (Wilhelm 1911, p. 107), for additional clarification.

By extensively citing Western religious, literary, and philosophical works in the paratexts, Wilhelm consciously expanded the dialogue space between Chinese philosophical thought and Western intellectual traditions. He neither criticized the *Daodejing* as a heretical work nor arbitrarily dissected Chinese philosophy by forcing it into Western philosophical systems. Instead, he demonstrated the compatibility of Chinese philosophy with 20th-century thought by utilizing Western concepts. Compared to earlier translators, Wilhelm's interpretive principles undoubtedly represent a significant advancement.

## 4. Conclusions

This article, taking Wilhelm's translation as an example, explores the significant role of paratexts in the translation of the *Daodejing* and the multiple interpretative paths that Wilhelm achieved. Utilizing the varied and easily comprehensible paratexts, Wilhelm's translation strikes a balance between popular appeal and academic rigor, obtaining unanimous approval from both general readers and academic scholars. More importantly, Wilhelm changed the long-standing interpretative practice of religiously metaphorizing the *Daodejing*. Previous translations merely treated the *Daodejing* as exotic material that could be cut or taken out of context, forcefully employing it to corroborate Christian thought. However, Wilhelm commenced from the Chinese thought itself and facilitated readers' understanding of the *Daodejing* through a combination of Chinese classics and Western literature. Since its publication, Wilhelm's translation has been reprinted nearly 30 times

and translated into 9 different languages, including English, French, and Italian. Moreover, 13 German translations of the *Daodejing* have referenced Wilhelm's version for retranslation (Tadd 2022a, pp. 146–147). With the canonization of Wilhelm's translation, theological interpretations of the *Daodejing* have decreased, while philosophical interpretations have significantly increased. For example, in Debon's interpretation of the *Daodejing*, he extensively references Chinese classis such as *Liezi* and *Zhuangzi*, evidencing a discernible influence from Wilhelm's framework (Debon 1961). In later editions of Strauss's translations, there is a notable reduction in theological content. Accompanying this change, extensive footnotes have been moved to post-textual interpretation, similarly influenced by Wilhelm's use of paratexts (Strauss 1987). Till today, the eminent Sinologist Eva Lüdi Kong, in her translation of *Xiyouji* 西遊記 (Journey to the West), employs extensive paratexts to facilitate comprehension. In her annotations, Kong often refers to Wilhelm's interpretations of Dao (Hu and Tan 2021), demonstrating the enduring influence of Wilhelm's methodology in contemporary Sinology.

However, the appropriate use of paratexts is only one of the reasons why Wilhelm's edition became a classic German translation. In addition to his own capabilities in translation, the cooperation with Diederichs publisher also contributed to the success of his translation. Renowned in Germany at that time, Diederichs publisher wielded considerable influence in the German-speaking world. Their great economic strength enabled Wilhelm's translation to maintain substantial circulation, even amidst the challenges of World War I and the post-war German economic crisis, securing its place in the German book market. Beyond translating Chinese classics, Wilhelm also published numerous papers and monographs on Chinese philosophy. His ever-increasing status in the fields of translation and sinology is another crucial factor in sustaining high interest in his *Daodejing* translation. Furthermore, Wilhelm organized over sixty public and academic lectures, various cultural seminars, themed exhibitions, and art lectures through the Frankfurt China Institute (das China-Institut Frankfurt am Main), which he founded (Leutner 2003, pp. 43–44). All these activities served as effective pathways to promote the acceptance and dissemination of his translation of the *Daodejing*. In addition to focusing on textual research, studying the global dissemination and canonization of the *Daodejing* from a sociological perspective could provide additional valuable insights for future studies.

**Author Contributions:** Conceptualization, X.L.; methodology, X.L.; formal analysis, X.L.; resources, X.L. and Y.T.; data curation, X.L.; writing—original draft preparation, X.L. and Y.T.; writing—review and editing, X.L. and Y.T.; project administration, Y.T.; supervision, Y.T.; funding acquisition, Y.T. All authors have read and agreed to the published version of the manuscript.

**Funding:** This research was funded by the Fundamental Research Funds for the Central Universities HUST: 2018WKZDPY007.

**Institutional Review Board Statement:** Not applicable.

**Informed Consent Statement:** Not applicable.

**Data Availability Statement:** Data are contained within the article.

**Conflicts of Interest:** The authors declare no conflict of interest.

## Notes

1  "Laozegetics" emerges from the Chinese study of *Laoxue* 老學, which means "the study of, doctrine of, school of, knowledge of, or field of study of Laozi the person or *Laozi* the text" (Tadd 2022b, p. 2). Laozegetics shifts the focus from seeking the original text and its original meaning to appreciating the hermeneutical and historical value of the various translations and interpretations on the *Daodejing*, including those in different cultures and languages.

2  Diederichs publisher was founded by Eugen Diederichs (1867–1930) in 1897. Since then, Diederichs has always led discussions on important social issues in Germany, dedicating himself to introducing the finest cultures of various nations into the German-speaking world. He can be considered one of the most significant figures in the German cultural sphere in the first half of the 20th century (Diederichs 2014, p. 8). In addition to the "Chinese Religion and Philosophy" series, Wilhelm also published a large number of academic monographs at this publishing house.

3    Before the appearance of Wilhelm's translation, there were eight full German translations of the *Daodejing* (Tadd 2022a, pp. 145–146, 156): 1. *Laò-Tsè's Taò Tĕ Kīng* (Victor von Strauss 1870); 2. *Lao-Tse Táo-Tĕ-King, der Weg zur Tugend* (Reinhold von Plaenckner, 1870); 3. *Taòtekking von Laòtsee* (Friedrich Wilhelm Noak, 1888); 4. *Theosophie in China. Betrachtungen über das Tao-Teh-King* (Franz Hartmann, 1897); 5. *Lao-tsï und seine Lehre* (Rudolf Dvorák, 1903); 6. *Die Bahn und der rechte Weg des Lao-Tse* (Alexander Ular, 1903); 7. *Des Morgenlandes grösste Weisheit. Laotse Tao Te King* (Joseph Kohler, 1908); 8. *Lao-tszes Buch vom höchsten Wesen und vom höchsten Gut* (Julius Grill, 1910).

4    *Yang* 陽 and *yin* 陰 are two significant concepts in Chinese philosophy, representing two fundamental forces in nature that are both opposing and interdependent. *Yang* symbolizes traits such as positivity, masculinity, daylight, and strength, while *yin* represents passivity, femininity, nighttime, and gentleness. These two principles are considered as the foundation for the existence and development of all things. Wilhelm's interpretation here referred to the content of Chapter 5 of *Yizhuan Xici* 易傳·系辭 (Commentary on the Attached Verbalizations of the *Yijing*), which states, "The interaction of *yin* and *yang* is called Dao".

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
