# Peer review of "A Paratext Perspective on the Translation of the Daodejing: An Example from the German Translation of Richard Wilhelm"

_religions, doi:10.3390/rel14121546_

Round 1
Reviewer 1 Report
Comments and Suggestions for Authors
A very solid and interesting article. Highly original and valuable contribution to the field. In terms of improvable aspects:
- a few minor issues related to consistency in stylistic conventions.
- the article could benefit from a better framing with respect to the field, however. Perhaps more engagement with scholars of Daoism who have worked on reception history or paratexts.
- there are some inklings of West vs. East orientalist language that can be excised or modified.
- Most importantly, the significance of paratexts and specifically, Wilhelm's choice of paratexts, could be elaborated upon. I felt the article could benefit from more analytical unpacking/theorizing at the end. Maybe only a few lines expanding on the work of others. (e.g. are Genette's ideas confirmed or contradicted by the findings; are the similar to the findings of others who have worked on Daoism and paratexts; is this use of paratexts consistent across sources and genres; did Wilhelm's use of paratexts impact later German or European translations, etc., etc.. ). In other words, shifting from micro questions to macro frames a bit.

Comments on the Quality of English LanguageThe English is generally fine, with the exception of some word choices (underlined in green) and certain formulations. The word "thought" should not be pluralized when referring to ideological or philosophical systems. Likewise, "paratexts" should be pluralized when referring to more than one paratext (i.e. as a group or category).
Finally, the conclusion is a bit less polished than the rest of the article. It deserves some extra attention
Author Response
【Please see the attachment】We would like to express our sincere gratitude for the time and effort you have devoted to reviewing our manuscript. We have thoroughly read and considered the valuable comments provided, and have made careful revisions to the manuscript accordingly. Please see the attachment to find the detailed responses below and the corresponding corrections in track changes in the resubmitted files.

Reviewer 2 Report
Comments and Suggestions for Authors
In my opinion this paper is well written and well structured, there is a consistency between the abstract, paragraphs, conclusions and the topic of analysis : "taking Wilhelm's rendition as an example, explores the significant function of paratext in the translation of the Daodejing and the multiple interpretative paths achieved by Wilhelm through the use of paratext". The issue is explored in detail and with clarity. I think the author knows the topic very well and this emerges from what is written.
My only suggestion is that perhaps the issues raised here could be explored in more detail:
" Wilhelm posits that Laozi's thoughts harbor an enduring value that transcends time, having now become a directional marker for European thoughts. He is keen to delineate between Taoist religion and philosophical Daoism, emphasizing that Laozi was not the founder of Taoist religion and that his philosophy was merely co-opted by Taoist religion." (p. 4).
I am referring to the relation to European thoughts and what does mean "was merely co-opted?". I think these are too important issues to be summarized in one sentence and then referred back to the essays published in the book as a bibliographical reference.
Author Response
Thank you for pointing this out. We agree with this comment and we fully understand your concerns. Our descriptions of these two important issues were overly simplistic. To more clearly articulate the aforementioned points, the revised content is as follows【See P.5, line199-215】:
Wilhelm posits that Laozi's thought harbor an enduring value that transcends time,“Laozi are increasingly starting to be picked up in Europe nowadays”(Wilhelm 1911, p. XIII). At the end of the 19th century and the beginning of the 20th century, Germany, amidst rapid industrialization and modernization, began to experience a modernity crisis of nihilism (Xu 2023, p. 62). This spiritual dilemma led many Germans to seek relief and guidance in the inner peace offered by Daoism. As Carl Gustav Jung expressed:“The spirit of the East is truly at the gates. Therefore, it seems to me that the realization of meaning, the seeking of Dao道, has already become a more collective phenomenon among us to a much greater extent than is generally thought”(Jung 1982, p. XVIII). Against the backdrop of Nietzsche’s prevailing philosophy and the waning of Christian faith, the Daoism to some extent took on the role of a new type of “salvation religion” for many Germans (Detering 2008, p. 27) . However, Wilhelm opposed the notion of viewing Daoism thought merely as a religious concept. He specifically cautioned readers: “What is commonly referred to as Daoist nowadays can be traced back to the animistic folk religion of ancient China”(Wilhelm 1911, p. XIV). Although the Daodejing is a foundational text of Daoist, it should not be solely regarded as a religious text. In reality, it comprises Laozi's exploration of the world's origins and a system of metaphysical philosophical thought.
-Additionally, we have rechecked the punctuation and the expressions in this article and rewritten long sentences that were unclear in their articulation. We have also reviewed and updated the references.